# Association between Primary Perioperative CEA Ratio, Tumor Site, and Overall Survival in Patients with Colorectal Cancer

**DOI:** 10.3390/jcm9123848

**Published:** 2020-11-27

**Authors:** Thomas A. Odeny, Nicole Farha, Hannah Hildebrandand, Jessica Allen, Wilfred Vazquez, Maximillian Martinez, Ravi Kumar Paluri, Anup Kasi

**Affiliations:** 1Department of Medicine, University of Missouri-Kansas City, Kansas City, MO 64108, USA; taodeny@gmail.com; 2Department of Medicine, University of Kansas Medical School, Kansas City, KS 66160, USA; nfarha2@kumc.edu (N.F.); hhildebrand@kumc.edu (H.H.); jallen19@kumc.edu (J.A.); wvazquez@kumc.edu (W.V.); mmartinez11@kumc.edu (M.M.); 3Department of Medicine, Wake Forest School of Medicine, Winston-Salem, NC 27101, USA; rpaluri@wakehealth.edu; 4Department of Medicine, Division of Medical Oncology, University of Kansas Cancer Center, 2650 Shawnee Mission Pkwy, Fairway, KS 66205, USA

**Keywords:** carcinoembryonic antigen, CEA, CEA ratio, colorectal tumor location

## Abstract

There are differences in the incidence, clinical presentation, molecular pathogenesis, and outcome of colorectal cancer (CRC) based on tumor location. Emerging research suggests that the perioperative carcinoembryonic antigen (CEA) ratio (post-op/pre-op CEA) is a prognostic factor for CRC patients. We aimed to determine the association between CEA ratio, tumor location, and overall survival (OS) among patients with CRC. We analyzed 427 patients who underwent resection for CRC at the University of Kansas Medical Center. After excluding those without pre- or post-operative CEA data, 207 patients were classified as either high (≥0.5) or low (<0.5) ratio. Primary outcomes were as follows: (1) OS stratified by CEA ratio; (2) OS stratified by tumor location; (3) OS stratified by tumor location among those with CEA elevation > 5 ng/mL at the time of recurrence. The Kaplan–Meier method was used to estimate survival rates. The median age was 62 years (inter-quartile range 51–71), 55% were male, 41% were smokers, 71% had left-sided tumors, the median pre-operative CEA was 3.1 ng/mL (inter-quartile range (IQR) 1.5–9.7), and 57% had a CEA ratio ≥0.5. The OS rates were 65.1% and 86.3% in patients with high versus low CEA ratios, respectively (log-rank *p*-value = 0.045). The OS rates were 64.4% and 77.3% in patients with right-sided vs. left-sided tumors, respectively (log-rank *p*-value = 0.5). Among patients with CEA levels greater than 5 at the time of recurrence, the OS rates were 42.9% and 43.4% in patients with right-sided vs. left-sided tumors, respectively (log-rank *p*-value = 0.7). There was a significantly higher survival among patients with low CEA ratios than among those with high CEA ratios. There was no difference in OS between left- versus right-sided tumors. Among patients with CEA elevation > 5 ng/mL at the time of recurrence, there was no difference in OS between left versus right-sided tumors. These findings warrant validation in a larger cohort as our sample size was limited.

## 1. Introduction

The outcomes of patients with colorectal cancer (CRC) differ based on tumor location [1]. For example, overall survival among patients who undergo curative resection for CRC differs according to the location of the tumor [2]. A recent systematic review found that left-sided colon cancer is associated with improved survival [3], suggesting that tumor sidedness could be an important prognostic factor. While postoperative serum carcinoembryonic antigen (CEA) level is routinely used as a marker for possible metastatic disease, the perioperative CEA ratio (post-op/pre-op CEA) may be an important emerging prognostic factor for CRC patients [4,5] and may be especially important in patients with high preoperative CEA levels [6]. To our knowledge, the association between perioperative CEA ratio and tumor location has not been previously established. We aimed to determine the association between CEA ratio, tumor location, and overall survival (OS) among patients with CRC.

## 2. Experimental Section

We conducted a retrospective cohort analysis of the effect of perioperative CEA ratio and the location of primary colorectal cancer on overall survival. Study subjects included all patients diagnosed with stage II and III biopsy-proven colorectal adenocarcinoma at the University of Kansas Medical Center (KUMC) between February 2008 and March 2018, who had their CEA levels checked during the course of treatment and follow up and who underwent definitive surgical resection with or without adjuvant chemotherapy and/or radiation. Study subjects had at least two years of follow up unless recurrence occurred earlier. Patients with other actively treated malignancies diagnosed in the last five years were excluded, except those with basal cell carcinoma of the skin, carcinoma in situ of the cervix, and ductal carcinoma in-situ of the breast.

Participant demographics, self-reported history of cigarette smoking, clinicopathological characteristics, and tumor recurrence information were collected using REDCap, a secure web-based electronic data capture software hosted at KUMC [7,8]. Patients who died of any other non-CRC cause that occurred before clinical recurrence of CRC were censored at the time of death.

We analyzed records of 427 patients with CRC who underwent resection at the University of Kansas Cancer Center between February 2008 and March 2018. CEA tests were obtained by standard electrochemiluminescence immunoassay as part of the routine care of these patients. We excluded those who did not have data on pre- or post-operative CEA levels. We calculated the CEA ratio as post-operative CEA levels/pre-operative CEA levels. Patients were classified as either high (≥0.5) or low (<0.5) CEA ratio based on previously published literature [9]. Subjects were further classified as having right-sided colon cancer if the primary tumor was located in the cecum, ascending colon, hepatic flexure, or transverse colon; and as having left-sided colon cancer if the tumor site was within the splenic flexure, descending colon, sigmoid colon, rectosigmoid junction, or rectum. We determined cancer recurrence with imaging evidence of distant disease.

Primary outcomes included overall survival (OS) stratified by CEA ratio, OS stratified by tumor location, and OS stratified by tumor location among those with CEA elevation >5 ng/mL at the time of recurrence. Overall survival was measured from the time of surgical resection to the time of death due to any cause. We used the Kaplan–Meier method to estimate survival rates. Cox proportional hazards models were used for multivariable analysis. Variables for inclusion in multivariable regression models were selected a priori based on known or suspected risk factors for survival. All tests were two-sided with a significance level of *p* < 0.05. Stata v13 was used to perform all analyses (StataCorp, College Station, TX, USA).

The retrospective study was reviewed by the University of Kansas Medical Center Human Research Protection Program. The type of review was a Flexible Institutional Review Board review. The study was eligible for Flexible IRB Review because it was deemed minimal risk and is not associated with any federal funding or support.

## 3. Results

We analyzed a total of 207 patients with median age of 62 years (inter-quartile range of 51–71). Of these, 55% were male, 41% were smokers, and 71% had left-sided tumors (Table 1). The median pre-operative CEA was 3.0 ng/mL (IQR 1.5–9.7), and the median post-operative CEA was 1.7 ng/mL (IQR 1.1–2.9). Overall, 57% had CEA ratio ≥ 0.5. (Table 1). The overall survival (OS) rates were 65.1% in patients with high CEA ratios and 86.3% in patients with low CEA ratios (log-rank *p*-value = 0.045) (Figure 1). The OS rates were 64.4% in patients with right CRC and 77.3% in patients with left CRC (log-rank *p*-value = 0.5) (Figure 2). Among patients with CEA levels greater than 5 at the time of recurrence, the OS rates were 42.9% in patients with right CRC and 43.4% in patients with left CRC (log-rank *p*-value = 0.7) (Figure 3).

When adjusted for smoking status and tumor stage, the effect of tumor location on survival was modified by perioperative CEA level (*p*-value for interaction term< 0.001). In the analysis stratified by tumor location, however, the numbers in the right-sided tumor group were too small to permit inferential analysis. Table 2 shows the effect of tumor location on OS, adjusting for perioperative CEA level, smoking status, and tumor stage without including an interaction term between perioperative CEA level and tumor location.

## 4. Discussion

In this study, we found that there was a significantly higher OS among patients with low perioperative CEA ratios compared to those with high perioperative CEA ratios. This finding lends support to the potential use of perioperative CEA levels as a prognostic factor for CRC patients [4,5]. Konishi et al. suggested that preoperative CEA levels may not provide useful information in the setting of normal post-operative CEA levels [4]. However, our results suggest that any level of post-operative CEA can be informative for prognosis if it is paired with pre-operative CEA levels as a ratio. The differing survival rates between patients with high versus low perioperative CEA ratios could potentially be a marker for how to allocate resources for surveillance. Patients with high perioperative ratios could potentially benefit from less frequent or less intense surveillance for recurrent disease, and those with low ratios may benefit from more frequent or more intense follow up. To confirm this, studies with longer follow up of these patients would be needed.

In secondary analyses, we also found that there was no difference in OS between left-sided CRC and right-sided CRC in our cohort. Specifically, there was no difference in OS between left- versus right-sided tumors among patients with post-operative CEA level > 5 at the time of recurrence. However, we found that the effect of tumor location on survival is modified by perioperative CEA levels, although our small sample size did not permit analyses stratified by CEA levels. This study is among the few to evaluate the association between perioperative CEA ratio and tumor location. While primary tumor location has important prognostic value for advanced colorectal cancer [10], our findings suggest that prognostic models that incorporate tumor sidedness may be enhanced further by inclusion of CEA ratio levels.

Our study was strengthened by having all the data from a single center with long and consistent follow up for the patients. This also ensured that CEA assays were all standardized and from the same lab, thus eliminating inter-lab discrepancies. We recognize the limitations of our study. First, the study did not have sufficient power to detect differences in survival among patients stratified by tumor location and CEA ratio. Such an analysis could potentially add precision to prognostic models for advanced colon cancer by taking into account both primary tumor location and perioperative CEA levels. The findings reported here, therefore, warrant validation in a larger cohort due to our limited sample size. Second, many potential study participants did not have their pre-operative or post-operative CEA levels measured. Therefore, our findings may reflect a biased sample that is not truly representative of target patients. Finally, the interpretation of our findings is limited by losses to follow up over time. Our survival analysis methods, however, account for losses to follow up and reduce bias by censoring patients at their last visit.

## 5. Conclusions

In conclusion, this study suggests that the perioperative CEA ratio may be an important emerging prognostic factor for CRC patients, and may exceed the current use of only post-operative CEA levels for prognostication. Patients may benefit from individualized follow up surveillance informed by perioperative CEA levels.

## Figures and Tables

**Figure 1 jcm-09-03848-f001:**
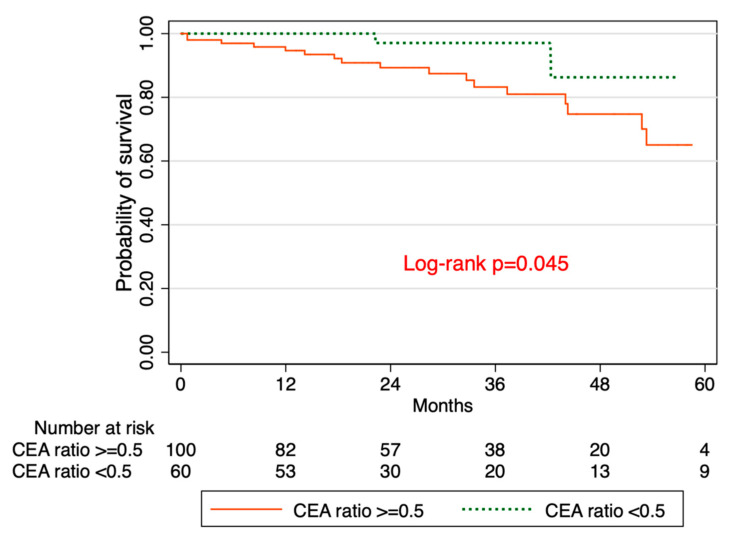
Kaplan–Meier survival based on carcinoembryonic antigen (CEA) ratio.

**Figure 2 jcm-09-03848-f002:**
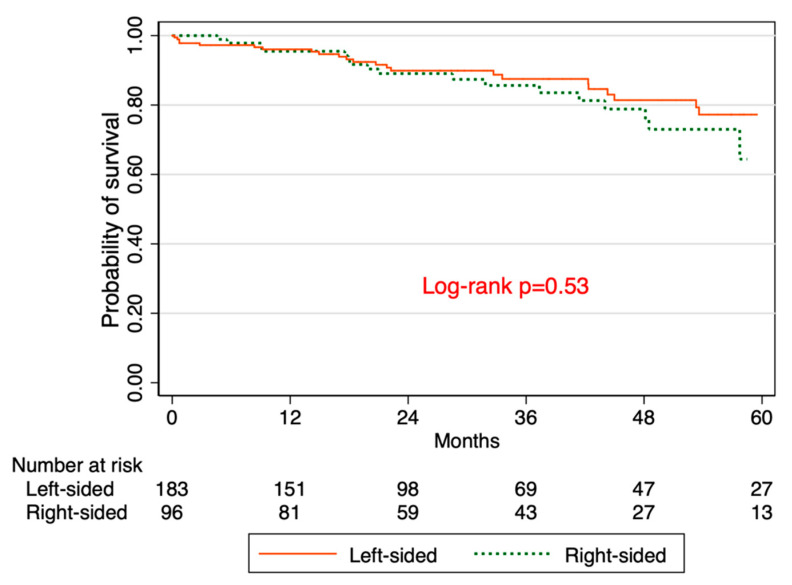
Kaplan–Meier survival based on tumor location.

**Figure 3 jcm-09-03848-f003:**
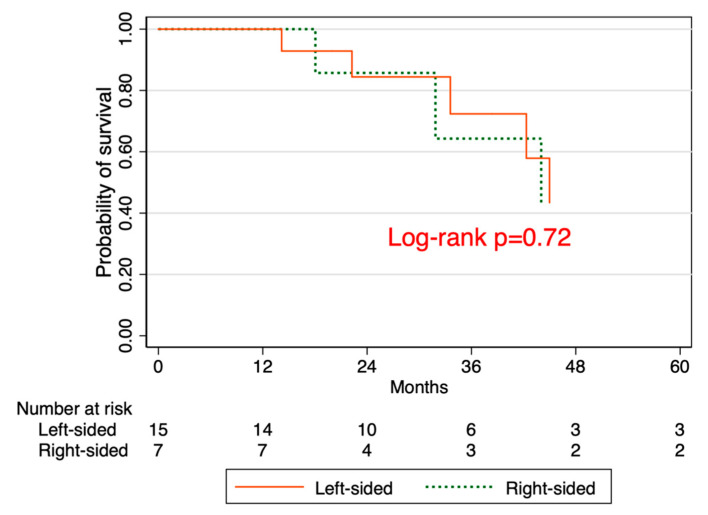
Kaplan–Meier survival based on tumor location among those with CEA >5 at the time of recurrence.

**Table 1 jcm-09-03848-t001:** Tumor location in patients categorized by age, sex, smoking status, CEA ratio, and tumor stage.

Characteristic	Tumor Location		Total *n* (%)
	Left-Sided *n* (%)	Right-Sided *n* (%)	
Age (years), median IQR	60 (48–68)	65 (52–73)	62 (51–71)
Sex			
Male	81 (82.7)	17 (17.4)	98 (55.4)
Female	45 (57.0)	34 (43.0)	79 (44.6)
Smoking			
No	69 (68.3)	32 (31.7)	95 (58.6)
Yes	50 (74.6)	17 (25.3)	67 (41.4)
CEA ratio			
<0.5	53 (69.7)	23 (30.3)	76 (42.7)
≥0.5	73 (71.6)	29 (28.4)	102 (57.3)
Tumor stage			
Stage 2	29 (63.0)	17 (37.0)	46 (36.2)
Stage 3	62 (76.5)	19 (23.5)	81 (63.8)

Note: Total numbers may not add up to 207 due to missing categorical data.

**Table 2 jcm-09-03848-t002:** The effect of tumor location on overall survival (OS), adjusting for perioperative CEA level, smoking status, and tumor stage.

Variable		Unadjusted	*p*-Value	Adjusted	*p*-Value
		HR (95% CI)		HR (95% CI)	
Tumor location	Right (*n* = 35)	Ref		Ref	
	Left (*n* = 88)	0.82 (0.45–1.50)	0.5	1.06 (0.28–3.96)	0.9
CEA ratio	<0.5 (*n* = 52)	Ref		Ref	
	≥0.5 (*n* = 71)	3.06 (0.97–9.60)	0.06	1.59 (0.45–5.34)	0.5
Smoking status	No (*n* = 79)	Ref		Ref	
	Yes (*n* = 44)	1.66 (0.95–2.91)	0.08	3.16 (0.97–10.32)	0.06
Tumor stage	Stage 2 (*n* = 43)	Ref		Ref	
	Stage 3 (*n* = 80)	1.61 (0.75–3.42)	0.2	1.46 (0.42–5.17)	0.6

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
