# Peer review of "Association between Primary Perioperative CEA Ratio, Tumor Site, and Overall Survival in Patients with Colorectal Cancer"

_jcm, 2020, doi:10.3390/jcm9123848_

Round 1

Reviewer 1 Report

Comments to JCM- 976448

CEA is a classical biomarker, commonly used for prediction of treatment response and surveillance of colorectal cancer (CRC) recurrence. However, a plethora of investigations considering the value of CEA as a monitoring parameter in CRC indicates that its potential has not been exhausted yet.

The study deals with association of perioperative CEA ratio with certain parameters such as patient survival, tumor location. Such an approach seems to be reasonable especially in relation to patients with high preoperative CEA level or normal postoperative CEA level.

In general, the study to a certain extent represents confirmatory work. Due to small groups of cases caused by missing categorical data the obtained results should be considered as preliminary (the authors admitted it themselves). However, the study may serve as a starting point for further investigations (e.g. value of the CEA ratio evaluation in clinical practice).

Comments:

The authors did not provide methodology of CEA measurement and details of sampling time are missing. Also a concentration unit of CEA should be given – as this information may be obvious for experts in this field, non-experts may not be familiar with it.

In the study rectal tumors (which represent over 25% of all CRC) have been included into the group of left-sided CRCs. Rectal and colon cancers differ in etiology, clinical management and outcomes. For this reason, not infrequently, rectal tumors are analyzed separately. Have the authors considered such an approach?

Please give the correct median preoperative CEA value; you wrote: “….3.0 (IQR 1.5-9.7)….” – line 23 and “…3.1 (IQR 1.5-9.7)…” – line 79.

The subtitle “3.1. Figures, Table……” (line 91) is dispensable.

Table 2 should be referred to in the text. As stated in Instruction for Authors tables/figures should be inserted into the main text close to their first citation.

Place the titles above the Tables and use a capital letter at the beginning of each title.

I suggest introducing some modification to the figures and figure captions:

In Figure 3 the title placed above the plot area is not clearly readable. Anyway, in my opinion, there is no need to repeat titles of the figures placing them above the plots.

Please remove “Fig….” from left upper corner of each figure.

Please use larger font size for log-rank p-values given on the plots to make them clearly readable.

Figures 1-3: Y-axis presents probability of survival in a range of 0.00 – 1.00 but not overall survival in [%].

Author Response

Dear Editorial Board of Journal of Clinical Medicine,

RE: RESPONSES TO REVIEWER COMMENTS

We would like to sincerely thank you for reviewing our manuscript, and for providing important comments. We have taken all of the referees’ comments into consideration and have revised the manuscript accordingly. Below, please find our point-by-point responses/

REVIEWER 1
Comment 1. The authors did not provide methodology of CEA measurement and details of sampling time are missing. Also a concentration unit of CEA should be given – as this information may be obvious for experts in this field, non-experts may not be familiar with it.

Response:
We appreciate the reviewer’s comment. We have revised the manuscript to include the concentration unit of the six CEA measurements where the concentration unit was missing. The remaining values relate to the CEA ratio. We have also provided the methodology for CEA measurement in the manuscript as follows: “CEA test was obtained by standard electrochemiluminescence immunoassay as part of routine care of these patients”

Comment 2. In the study rectal tumors (which represent over 25% of all CRC) have been included into the group of left-sided CRCs. Rectal and colon cancers differ in etiology, clinical management and outcomes. For this reason, not infrequently, rectal tumors are analyzed separately. Have the authors considered such an approach?

Response:

Thank you for this comment. We agree that rectal and colon cancers differ in etiology, clinical management, and outcomes. However, we did analyze rectal tumors separately due to the small sample size in our dataset. We acknowledge this as a limitation of the study.

Comment 3. Please give the correct median preoperative CEA value; you wrote: “….3.0 (IQR 1.5-9.7)….” – line 23 and “…3.1 (IQR 1.5-9.7)…” – line 79.

Response:

Thank you for pointing this out. We have provided the correct median preoperative CEA values as follows:

Line 23: “…median pre-operative CEA was 3.0 ng/mL (IQR 1.5-9.7).”

Line 79: “…median pre-operative CEA was 3.0 ng/mL (IQR 1.5-9.7).”

Comment 4. The subtitle “3.1. Figures, Table……” (line 91) is dispensable.

Response:

We have removed the figure subtitles as suggested.

Comment 5: Table 2 should be referred to in the text. As stated in Instruction for Authors tables/figures should be inserted into the main text close to their first citation.

Place the titles above the Tables and use a capital letter at the beginning of each title.

Response: We appreciate this comment. Table 2 is indeed referred to in the text (line 89). We have inserted the tables/figures into the main text close to their first citation, which is the results section. We have revised the tables by placing all titles above the tables and we have also used a capital letter at the beginning of each title.

Comment 6: I suggest introducing some modification to the figures and figure captions:

In Figure 3 the title placed above the plot area is not clearly readable. Anyway, in my opinion, there is no need to repeat titles of the figures placing them above the plots.

Please remove “Fig….” from left upper corner of each figure.

Please use larger font size for log-rank p-values given on the plots to make them clearly readable.

Figures 1-3: Y-axis presents probability of survival in a range of 0.00 – 1.00 but not overall survival in [%].

Response: We appreciate and agree with all these comments. We have revised Figure 3 by removing the title in the figure and kept the title outside the figure. We have removed “Fig…” from the left upper corner of each figure. We have used a larger font size for log-rank p-values to make them clearly readable. We have also modified the Y-axis labels to read “Probability of survival.”

Once again, thank you for your consideration of our manuscript for publication in JCM the Journal of Clinical Medicine.

Sincerely,

Anup Kasi
On behalf of the authors.

Reviewer 2 Report

The authors performed a retrospective study of 207 patients over the course of a decade to assess whether the CEA ratio (post/pre) was associated with differences in survival.

The paper was clear and succinct and the tables were informative.

I have a couple of questions/comments:

  • What was the range of CEA levels preop? Mean, median?
  • What was the range of CEA levels postop? Mean? Median? Did all patients have normal CEA levels postop? At what time was the first postop CEA level taken?
  • Did all patients receive R0 resections? Adequate lymph node harvests?
  • Which patients received adjuvant chemo and did this affect their OS?

Thank you for the opportunity to review your work.

Author Response

Dear Editorial Board of Journal of Clinical Medicine,

RE: RESPONSES TO REVIEWER COMMENTS

We would like to sincerely thank you for reviewing our manuscript, and for providing important comments. We have taken all of the referees’ comments into consideration and have revised the manuscript accordingly. Below, please find our point-by-point responses/

REVIEWER #2

Comments:

  • What was the range of CEA levels preop? Mean, median?
  • What was the range of CEA levels postop? Mean? Median?
  • Did all patients have normal CEA levels postop? At what time was the first postop CEA level taken?
  • Did all patients receive R0 resections? Adequate lymph node harvests?
  • Which patients received adjuvant chemo and did this affect their OS?

Responses: The means and ranges of CEA levels were:

Pre-op: mean 9.94 (range 0.4-330)
Post-op: mean 4.8 (range 0.3-212)

The values for pre-op and post-op CEA levels were not normally distributed so we chose to present in the manuscript only the median and IQR values. As can be seen by the distribution of post-op CEA levels, not all patients had normal CEA levels post-op. The first post-op CEA level was taken in the range of 4-6 weeks after surgery. We did not collect data on whether all patients received R0 resections or whether lymph node harvests were adequate. We appreciate these comments and will incorporate these data in future analyses.

Once again, thank you for your consideration of our manuscript for publication in JCM the Journal of Clinical Medicine.

Sincerely,

Anup Kasi
On behalf of the authors.

Reviewer 3 Report

1) Please include IQR in line 23 so that readers can know that it will be used as a abbreviation.

2) Why does the exclusion criteria excludes all active malignancies but not endometrial cancer grade 1/stage 1? There is some data published in few reports that CEA can be used as a tumor marker for endometrial cancer patients also. why cant be that excluded?

3) Were all the patients retrospectively consented for data collection?

4) In Table 1, the total number of subjects included in the study (207) is not the same when we add the N for the variables. For example. 98 male and 78 females were in the study so total of 176 patients. Is there some data missing? It looks like the N varied for all the variables. If so how will be the statistical analysis be accurate?

5) whats the reason for choosing post operative CEA of 5 at the time of recurrence when comparing the survival based on tumor location? why cant a higher number like above 15 be used as in the prior publications it was noted that CEA between 5-15 have high incidence of false positives and above 15 false positives are rare.

6) "This study is among the first to evaluate the 137 association between perioperative CEA ratio and tumor location" please explain this in detail as my understanding is that the sample size is too small to come to this inference. 

7) Line 141 CE assays need to be corrected as CEA assays. 

Author Response

Dear Editorial Board of Journal of Clinical Medicine,

RE: RESPONSES TO REVIEWER COMMENTS

We would like to sincerely thank you for reviewing our manuscript, and for providing important comments. We have taken all of the referees’ comments into consideration and have revised the manuscript accordingly. Below, please find our point-by-point responses/

REVIEW #3

Comment 1: Please include IQR in line 23 so that readers can know that it will be used as an abbreviation.

Response: Thank you for this comment. We have written out IQR in full (inter-quartile range) in line 23 so that readers can know it will be used as an abbreviation.

Comment 2: Why does the exclusion criteria excludes all active malignancies but not endometrial cancer grade 1/stage 1? There is some data published in few reports that CEA can be used as a tumor marker for endometrial cancer patients also. why cant be that excluded?

Response: We completely agree with this comment. We reviewed our dataset and none of the patients had concomitant stage 1 endometrial cancer. So we have excluded endometrial cancer grade 1/stage 1 per reviewer suggestion.

Comment 3: Were all the patients retrospectively consented for data collection?

Response: Thank you for asking this question. The consent requirement was waived by our IRB as this was retrospective chart review study. The study was reviewed by the University of Kansas Medical Center Human Research Protection Program. The type of review was a Flexible Institutional Review Board review. The study was eligible for Flexible IRB Review because it was deemed minimal risk and is not associated with any federal funding or support. We have indicated the same in the manuscript.

Comment 4: In Table 1, the total number of subjects included in the study (207) is not the same when we add the N for the variables. For example. 98 male and 78 females were in the study so total of 176 patients. Is there some data missing? It looks like the N varied for all the variables. If so how will be the statistical analysis be accurate?

Response: Thank you for this comment. The discrepancies in the N are indeed due to missing data. We have pointed this out in a note appended to the title for Table 1 (“Total numbers may not add up to 207 due to missing categorical data”). The missing data did not preclude statistical analysis but may be a source of bias. We considered performing multiple imputation but decided against it as we could not justify the “missing completely at random” assumption. As noted in the conclusion, the interpretation of our findings is limited by missing data (including missing at baseline & losses to follow up over time).

Comment 5: What’s the reason for choosing post operative CEA of 5 at the time of recurrence when comparing the survival based on tumor location? why cant a higher number like above 15 be used as in the prior publications it was noted that CEA between 5-15 have high incidence of false positives and above 15 false positives are rare.

Response: Thank you for this comment. The selection of CEA level of 5 was based on prior publications that used this cut-off (REF: Xie HL, Gong YZ, Kuang JA, Gao F, Tang SY, Gan JL. The prognostic value of the postoperative serum CEA levels/preoperative serum CEA levels ratio in colorectal cancer patients with high preoperative serum CEA levels. Cancer Manag Res. 2019;11:7499-7511).

Comment 6: This study is among the first to evaluate the association between perioperative CEA ratio and tumor location" please explain this in detail as my understanding is that the sample size is too small to come to this inference.

Response: Thank you for this comment. This statement only refers to the fact that few studies have evaluated this association, and that our present study is among these few. To clarify, we have replaced the word “first” with “few.”

Comment 7: Line 141 CE assays need to be corrected as CEA assays.

Response: Thank you for the comment. We have corrected this to read “CEA.”

Once again, thank you for your consideration of our manuscript for publication in JCM the Journal of Clinical Medicine.

Sincerely,

Anup Kasi
On behalf of the authors.